Development of a prediction model for lower limb deep vein thrombosis in critically ill patients after intracranial hemorrhage at high altitude: a retrospective study

Ciren Zhuoma 1
Fu Jianlei 1
Lin Guoying 1
Li Qianwei 1
Wang Bin wang-bin@pku.edu.cn binbisheng@163.com 2
1 Department of Intensive Care Medicine, Xizang Autonomous Region People’s Hospital , Lhasa , Xizang , China
2 Department of Critical Care Medicine, Peking University People’s Hospital , Beijing , China
Anson Lesley
Electronic publication date: 2025 Nov 13
Publication date: 2025
Volume: 13
Electronic Location ID: e20245
Received 2025 Mar 14; Accepted 2025 Sep 25
Copyright: ©2025 et al.
Copyright year: 2025
Copyright holder: et al.
License: This is an open access article distributed under the terms of the Creative Commons Attribution License, which permits unrestricted use, distribution, reproduction and adaptation in any medium and for any purpose provided that it is properly attributed. For attribution, the original author(s), title, publication source (PeerJ) and either DOI or URL of the article must be cited.
License URL: https://creativecommons.org/licenses/by/4.0/

Keywords: High-altitude area, Cerebral hemorrhage, Lower limb deep vein thrombosis, Risk factors, Prediction model

Funding: The Xizang Autonomous Region Natural Science Foundation Group Medical Aid Project XZ2024ZR-ZY007 (Z) This work was supported by the Xizang Autonomous Region Natural Science Foundation Group Medical Aid Project (XZ2024ZR-ZY007 (Z)). The funders had no role in study design, data collection and analysis, decision to publish, or preparation of the manuscript.

==============================
Objective

To analyze the risk factors for lower limb deep vein thrombosis (LDVT) in critically ill patients admitted to the intensive care unit (ICU) following intracranial hemorrhage at high altitude, and to establish a predictive model.

Methods

A retrospective analysis was conducted on 359 patients who underwent surgery for intracranial hemorrhage and were admitted to the ICU of Xizang Autonomous Region People’s Hospital between August 1, 2021, and December 31, 2023. Patients were categorized into an LDVT group (n = 86) and a non-LDVT group (n = 273) based on the occurrence of LDVT during their ICU stay. Demographic data, comorbidities, laboratory results, and treatment approaches were compared between the groups. Independent risk factors were identified through univariate and multivariate logistic regression analyses. A risk prediction model was developed using R software, and its performance was internally validated.

Results

Compared to the non-LDVT group, patients in the LDVT group had significantly longer ICU and overall hospital stays (p < 0.05). Multivariate logistic regression revealed hypoalbuminemia and elevated D-dimer levels as independent risk factors for LDVT, while aboriginal residency in high-altitude areas (≥ 4,500 m) was identified as a protective factor. A nomogram incorporating these variables was constructed. Internal validation demonstrated strong agreement between predicted and observed outcomes, with the area under the receiver operating characteristic (ROC) curve reaching 0.815 (95% CI [0.761–0.870]). The Hosmer-Lemeshow test indicated a good model fit (p = 0.088).

Conclusion

LDVT significantly prolongs ICU and hospital stays in critically ill patients. Hypoalbuminemia and elevated D-dimer levels are independent risk factors for LDVT, whereas aboriginal residency at high altitudes (≥ 4,500 m) serves as a protective factor. The developed risk prediction model shows strong predictive performance.

Introduction

Intracranial hemorrhage is a common and critical condition encountered in neurosurgery and intensive care units (ICUs), characterized by acute onset, rapid disease progression, and high mortality rates (DeLago Jr et al., 2022). Deep vein thrombosis (DVT) is a pathological condition caused by abnormal coagulation within the deep veins, leading to impaired venous return (Wolf et al., 2024). Studies have shown that DVT has the highest incidence among ICU patients, with reported rates ranging from 8% to 40%, and that over 80% of DVT cases occur in the lower limbs. Lower limb deep vein thrombosis (LDVT) is a frequent complication in postoperative patients, particularly in those with intracranial hemorrhage, who are more susceptible to LDVT due to limb dysfunction, reduced muscle strength, prolonged bed rest, and decreased venous blood flow. Pulmonary embolism and other thromboembolic complications secondary to LDVT can severely impact patient prognosis and quality of life, and may even result in death (Gil-Diaz et al., 2024).

High-altitude regions, defined as areas above 3,000 m above sea level (Mallet et al., 2021), are characterized by reduced oxygen partial pressure compared to sea level. Chronic hypoxia at these elevations can cause capillary swelling, increased permeability, and hemoconcentration, elevating the risk of intracranial hemorrhage (Rosenberg et al., 2024). Moreover, hypoxic conditions alter coagulation processes, promoting thrombosis (Damodar et al., 2018) by inducing endothelial inflammation, reducing cellular activity, enhancing procoagulant factors, and impairing coagulation function. Individuals residing above 3,000 m for over 11 months face a 30-fold higher risk of thromboembolic events (Anand et al., 2001). Consequently, patients with intracranial hemorrhage at high altitudes have a greater likelihood of developing LDVT than those at lower elevations, with distinct influencing factors and mechanisms.

At present, the internationally recognized DVT assessment scales include the Caprini scale and the Padua model (Hayssen et al., 2022). These tools were developed based on populations at low altitudes and have been shown to be effective in perioperative, general inpatient, and outpatient populations. However, the predictive performance of these tools for LDVT at high altitudes has not been validated, and there is currently no effective risk prediction model for high altitudes (Trunk, Rondina & Kaplan, 2019).

This study was conducted at the Xizang Autonomous Region People’s Hospital, which is located in Lhasa, 3,658 m above sea level, and is the stroke center of the Xizang Autonomous Region. This study aimed to develop a standardized scale for LDVT in critically ill patients in high-altitude areas to promote early diagnosis, targeted prevention, and standardized treatment plans, thereby improving the prognosis of patients in these areas.

Materials and methods

Study population

This retrospective study included patients who were admitted to the ICU of the Xizang Autonomous Region People’s Hospital for intracranial hemorrhage and underwent surgical treatment between August 1, 2021, and December 31, 2023. The study was approved by the hospital’s Medical Ethics Committee (Approval No. ME-TBHP-24-KJ-051). The consent was obtained for the use of retrospective data, and informed consent was waived due to the study’s retrospective design.

Intracranial hemorrhage types included cerebral hemorrhage, intraventricular hemorrhage, subarachnoid hemorrhage, subdural hemorrhage, and extradural hemorrhage. All patients with either primary or secondary intracranial hemorrhage who required surgical intervention and were transferred to the ICU postoperatively were included in the study. Surgical indications for each patient were determined by the attending surgeons based on their clinical judgment. The neurosurgical team at our hospital functions as a large, integrated unit, ensuring consistency in the application of surgical criteria across all team members. Detailed surgical indications are provided in the supplementary documents.

The inclusion criteria were as follows: (1) admission between August 1, 2021, and December 31, 2023, for intracranial hemorrhage requiring surgical intervention; (2) age ≥18 years; (3) Acute Physiology and Chronic Health Evaluation (APACHE) II score ≥10; and (4) ICU stay duration ≥24 h.

The exclusion criteria were: (1) patients diagnosed with LDVT at the time of ICU admission and (2) patients with incomplete medical records.

A total of 393 patients were screened for eligibility. Of these, 34 were excluded (nine due to preexisting LDVT and 25 due to incomplete data). Ultimately, 359 patients met the criteria and were included in the study (Fig. 1).

Study methods

Patient demographic data, clinical characteristics, laboratory values, and treatment modalities were retrospectively collected and analyzed. Demographic variables included sex, age, aboriginal residence at high altitudes (≥4,500 m), occupation (peasant or herdsman), smoking status, drinking status, and ethnicity (Tibetan). ICU admission status and comorbidities, such as the Glasgow Coma Scale (GCS) score at admission, APACHE II score, history of polytrauma, previous lower limb surgery, hypoalbuminemia, hypertension, and diabetes, were also recorded.

Figure 1 Patient selection flowchart.

Laboratory parameters included white blood cell count, platelet count, hemoglobin level, prothrombin time, activated partial thromboplastin time, fibrinogen level, and D-dimer level. Data on medications and blood transfusions, such as mannitol, tranexamic acid, glucocorticoids, vasoactive agents, red blood cells, plasma, platelets, fibrinogen, prothrombin complex concentrate, and hemocoagulase, were also collected. Laboratory test results reflected the values measured on the day of LDVT detection in affected patients. Medication and transfusion data were recorded prior to the occurrence of LDVT. All patients underwent laboratory testing upon hospital admission and again upon ICU admission. During the first week of ICU stay, tests were conducted every morning, and thereafter, every other day.

Daily lower-limb venous ultrasound examinations were performed on all ICU patients by trained medical personnel, and the results were recorded. Two qualified physicians conducted these ultrasound examinations every morning, screening both lower limbs.

Patients were categorized into the LDVT group or the non-LDVT group based on whether LDVT occurred during the ICU stay. Differences between the two groups were analyzed to identify independent risk factors for LDVT. A risk prediction model was subsequently developed and internally validated to assess its predictive performance.

Statistical analysis

Statistical analyses were performed using SPSS version 23.0. Normally distributed data are presented as means ± standard deviations and were analyzed using the independent samples t-test. Non-normally distributed data are presented as medians with interquartile ranges (IQR), and comparisons between groups were performed using nonparametric tests. Categorical variables are expressed as numbers (percentages) and were compared between groups using the chi-square test or Fisher’s exact test, as appropriate. Multivariate logistic regression analysis was conducted to identify independent risk factors for LDVT. R software (version 4.0.3) was used to construct the nomogram model. A two-tailed p-value ≤ 0.05 was considered statistically significant.

Results

Comparison of prognostic indices between the LDVT group and the non-LDVT group

Among the 359 high-altitude patients who underwent surgery for intracranial hemorrhage and were admitted to the ICU, 86 patients (23.96%) developed LDVT. Both ICU stay duration and total hospital stay were significantly longer in the LDVT group compared to the non-LDVT group (p < 0.05). Although the difference in mortality rates between the two groups was not statistically significant, the LDVT group had a mortality rate of 6.98%, compared to 3.66% in the non-LDVT group (Table 1).

Table 1 Comparison of prognostic indexes between LDVT group and non-LDVT group.

Variables	LDVT group
(n = 86)	non-LDVT group
(n = 273)	Statistic	p-value	
LOS of ICU, d, M (Q1, Q3)	15.5 (12, 24)	12 (9, 17)	Z =  − 4.415	<0.001	
Duration of hospital stay, d, M (Q1, Q3)	37 (24,60)	27 (6,44)	Z =  − 3.000	0.003	
ICU mortality, n (%)	6 (6.98)	10 (3.66)	χ2 = 0.998	0.318	
Notes.

Abbreviations LOS length of stay

ICU intensive care unit

Comparison of baseline data between the LDVT group and non-LDVT group

Univariate analysis revealed statistically significant differences between the two groups in terms of age, aboriginal residence at high altitudes (≥4,500 m), occupation as herdsman, hypoalbuminemia, thrombocytosis, elevated D-dimer levels, use of tranexamic acid, and plasma transfusion (p < 0.05, Table 2).

Table 2 Comparison of baseline data between LDVT group and non-LDVT group.

Variable	LDVT group
(n = 86)	Non-LDVT group
(n = 273)	Statistic	p-value	
Gender, n			χ2 = 0.541	0.462	
Male	52	177			
Female	34	96			
Age, year, M (Q1, Q3)	55.5 (45.75, 66)	50 (38, 59)	Z =  − 3.180	0.001	
Aborigines of high altitude areas (≥4,500m), n(%)	2 (60.47)	84 (64.83)	χ2 = 26.281	<0.001	
Peasant, n (%)	41 (47.67)	108 (39.56)	χ2 = 1.643	0.200	
Herdsman, n (%)	6 (6.98)	64 (23.44)	χ2 = 11.553	0.001	
Tibetan, n (%)	72 (83.72)	217 (79.49)	χ2 = 0.563	0.453	
Apache II score, score, M (Q1, Q3)	16 (12, 19)	16 (12, 19)	Z =  − 0.237	0.813	
GCS score, score, M (Q1, Q3)	8.5 (5, 15)	10 (5, 14)	Z =  − 0.584	0.559	
Polytrauma, n (%)	11 (12.79)	38 (13.92)	χ2 = 0.071	0.790	
Lower limb surgery, n (%)	0 (0.00)	10 (36.63)	χ2 = 2.040	0.153	
Hypoalbuminemia, n (%)	80 (93.02)	170 (62.27)	χ2 = 28.897	<0.001	
Hypertension, n (%)	40 (46.51)	114 (41.76)	χ2 = 0.564	0.453	
Diabetes, n (%)	1 (1.16)	5 (1.83)	χ2 = 0.184	0.668	
Smoker, n (%)	23 (26.74)	73 (26.74)	χ2 = 0.000	0.999	
Drinker, n (%)	26 (30.23)	74 (27.11)	χ2 = 0.318	0.573	
Laboratory values					
WBC, ×109 /L, M (Q1, Q3)	10.2 (7.95,12.1)	10.5 (8,13.1)	Z =  − 0.581	0.562	
PLT, ×109 /L, M (Q1, Q3)	143 (78, 223.75)	198 (147.75, 270.25)	Z =  − 3.381	0.001	
HGB, g/L, M (Q1, Q3)	122.5 (100.25,157)	131.50 (107,155)	Z =  − 0.822	0.411	
PT, s, M (Q1, Q3)	13.8 (12.98, 15.2)	14.1 (13, 15.12)	Z =  − 1.133	0.257	
APTT, s, M (Q1, Q3)	37.4 (33.82, 43.97)	38.15 (34.85, 42.15)	Z =  − 0.785	0.432	
FIB, g/L, M (Q1, Q3)	5.25 (3.53, 6.52)	5.85 (4.52, 6.78)	Z =  − 1.478	0.139	
D-dimer, mg/L, M (Q1, Q3)	8.25 (4.2, 16.34)	3.99 (2.4, 7.82)	Z =  − 5.705	<0.001	
Treatment measures					
Mannitol, n (%)	86 (100.00)	272 (99.63)	χ2 = 0.549	0.459	
Tranexamic acid, n (%)	29 (33.72)	126 (46.15)	χ2 = 4.248	0.039	
Glucocorticoids, n (%)	24 (27.91)	67 (24.54)	χ2 = 0.288	0.591	
Vasoactive agent, n (%)	38 (44.19)	144 (52.75)	χ2 = 2.093	0.148	
Red blood cell transfusion, n (%)	15 (17.44)	35 (12.82)	χ2 = 1.165	0.280	
Plasma transfusion, n (%)	18 (20.93)	29 (10.62)	χ2 = 6.107	0.013	
Platelet infusion, n (%)	4 (4.65)	4 (1.47)	χ2 = 1.760	0.185	
Hemocoagulase, n (%)	50 (58.14)	179 (65.57)	χ2 = 1.798	0.180	
Prothrombin complex infusion, n (%)	3 (3.49)	5 (1.83)	χ2 = 0.234	0.628	
Human fibrinogen infusion, n (%)	8 (9.30)	38 (13.92)	χ2 = 0.723	0.395	
Notes.

Abbreviations GCS Glasgow coma scale

WBC white blood cell

PLT platelet counts

HGB hemoglobin

Apache II Acute Physiology and Chronic Health Evaluation II

PT prothrombin time

APTT active partial thromboplastin time

Logistic regression analysis of LDVT risk factors

Significant factors identified in the univariate analysis were included as covariates in the multivariate logistic regression to determine independent risk factors for LDVT. The results showed that hypoalbuminemia (odds ratio (OR) = 6.109, 95% confidence interval (CI) [2.229–16.739], p < 0.001) and elevated D-dimer levels (OR = 1.028, 95% CI [1.000–1.055], p = 0.047) were independent risk factors for LDVT occurrence. In contrast, aboriginal residence at high altitudes (≥4,500 m) (OR = 0.052, 95% CI [0.005–0.430], P = 0.006) was identified as a protective factor against LDVT (Table 3).

Table 3 Logistic regression analysis of LDVT risk factors.

Variable	SE	Z	OR (95%CI)	p	
Age	0.011	3.163	1.020 (0.998–1.042)	0.075	
Aborigines of high altitude areas (≥4,500 m)	1.076	7.539	0.052 (0.006–0.430)	0.006	
Herdsman	0.618	0.216	0.750 (0.224–2.520)	0.642	
Hypoalbuminemia	0.514	12.381	6.109 (2.229–16.739)	<0.001	
PLT	0.002	1.601	0.998 (0.994–1.001)	0.206	
D-dimer	0.014	3.954	1.028 (1.000–1.055)	0.047	
Tranexamic acid	0.327	1.833	0.642 (0.339–1.219)	0.176	
Plasma transfusion	0.422	0.055	0.906 (0.396–2.070)	0.814	
Notes.

Abbreviations PLT platelet counts

SE standard error

OR Odd ratio

CI confidence interval

Construction and validation of the prediction model

Using R software (version 4.0.3), a nomogram prediction model for LDVT was constructed (Fig. 2). Internal validation with 1,000 bootstrap samples demonstrated that the area under the ROC curve (AUC) was 0.815 (95% CI [0.761–0.8697]) (Fig. 3). The calibration curve indicated good agreement between the predicted and observed outcomes (Fig. 4). Decision curve analysis (DCA) showed that the model had clinical utility with threshold probabilities ranging from 0.01 to 0.89 (Fig. 5). These results suggest that the prediction model performs well in assessing the risk of LDVT in patients with high-altitude intracranial hemorrhage.

Figure 2 Nomogram prediction model.

Figure 3 ROC curve of LDVT prediction model.

Figure 4 Correction curve of LDVT prediction model.

Figure 5 Decision curve analysis (DCA) for the prediction model.

Discussion

LDVT is a common complication in patients with intracranial hemorrhage, especially those in the ICU. Compared with patients undergoing other surgical procedures, patients with long-term bed rest and a longer recovery period are more likely to develop LDVT (Geerts et al., 2004). Chu et al. (2021) analyzed the clinical data of 848 critically ill patients with intracranial hemorrhage in the United States and reported that the incidence of venous thrombosis was 8.14%. In the present study, the incidence of LDVT among 359 high-altitude patients was 23.96%, significantly higher than the rate reported by Chu et al., likely due to the inclusion of a high-altitude population in this study.

This study identified hypoalbuminemia and elevated D-dimer levels as independent risk factors for LDVT, while being a aboriginal resident of high-altitude areas (≥4,500 m) was found to be a protective factor.

Hypoalbuminemia and LDVT risk

Albumin accounts for approximately 80% of plasma colloid osmotic pressure and plays a key role in regulating the dynamic balance of water between tissues and blood vessels, thereby increasing blood volume and maintaining stable colloid osmotic pressure (Sleep, 2015). When serum albumin concentration decreases, plasma colloid osmotic pressure is reduced, leading to the leakage of fluid from the vessels into the interstitial spaces. This process increases blood viscosity, slows blood flow velocity, and consequently raises the risk of thrombosis. In addition, hypoalbuminemia reduces plasma antithrombin activity, further elevating the risk of LDVT (Vincent et al., 2014).

In patients with intracranial hemorrhage, surgical trauma and acute stress result in the consumption of antithrombin III (Brangenberg, Bodensohn & Burger, 1997). Moreover, vascular endothelial injury and increased capillary permeability in hypoalbuminemic patients further lower effective antithrombin levels (Brangenberg, Bodensohn & Burger, 1997). Together, these factors contribute to a heightened risk of DVT in this patient population.

Elevated D-dimer levels as a risk factor for LDVT

This study identified D-dimer as an independent risk factor for LDVT. As a specific degradation product of cross-linked fibrin, D-dimer reflects in vivo coagulation and fibrinolysis, with elevated levels indicating active thrombosis and fibrinolytic activity (Zhang et al., 2019). It is widely recognized as a sensitive biomarker for thrombosis prediction, validated by numerous studies (Chopard, Albertsen & Piazza, 2020; Farzaneh et al., 2023). Delgado et al. (2006) also reported a correlation between elevated D-dimer levels and increased intracranial hemorrhage incidence, suggesting an association with early neurological deterioration and poor prognosis. Prolonged bed rest and a higher complication rate in these patients further support D-dimer’s predictive value for thrombosis. Its utility in DVT screening among ischemic and hemorrhagic stroke patients has been confirmed by multiple studies (Johansson et al., 2018; Chen, Zhang & Liu, 2023).

Aborigines of high-altitude areas as a protective factor of LDVT

High-altitude environments, characterized by hypoxia, low atmospheric pressure, and dry climates, are generally associated with an increased thrombotic risk (Zhu et al., 2021). However, this study found that aborigines living at altitudes ≥4,500 m were less likely to develop LDVT, suggesting that these populations may have developed intrinsic protective adaptations against thrombosis. Previous research indicates that high-altitude residents often exhibit polycythemia as a response to chronic hypoxia, with hematocrit levels increasing in proportion to altitude. To compensate, their fibrinolytic activity is enhanced, consumption of coagulation factors and fibrinogen rises, platelet adhesiveness decreases, and erythrocyte aggregation diminishes, collectively reducing the tendency toward hypercoagulability (Clivillé et al., 1998; Gao et al., 2023; Li et al., 2021).

In the present study, the median hemoglobin levels were relatively normal (122.5 g/L in the LDVT group and 131.5 g/L in the non-LDVT group), likely due to blood loss, fluid resuscitation, or other interventions upon ICU admission that may have reduced procoagulant factors. Despite these adjustments, the inherent physiological adaptations of high-altitude aborigines appeared to play a more decisive role in lowering LDVT risk. Notably, all patients residing above 4,500 m were Tibetan aborigines from the highest-altitude regions (Nagqu and Ngari prefectures). In contrast, the lower-altitude group comprised individuals of diverse ethnic backgrounds, including Han people and transient visitors, who may lack such adaptive mechanisms.

Prediction model of ldvt performance and implications

The Caprini score is a commonly used thrombotic risk assessment tool that is not specifically designed for high-altitude populations and includes many factors that are not related to the high-altitude environment (Golemi et al., 2019). Although the model in this study was relatively simple (containing only three factors: hypoalbuminemia, D-dimer, and high altitude), its predictive performance was strong, with an AUC of 0.815. The finding that high altitude reduces thrombotic risk is consistent with clinical experience in plateau medicine and emphasizes the specificity of this model for patients on the Qinghai-Xizang Plateau.

In addition to the factors included in this study, other potential risk factors, such as mobility status and prophylactic anticoagulant use, were not considered. However, in our center, patients remain largely immobilized during their ICU stay prior to discharge, minimizing variability in mobility status. Furthermore, as the study population consisted of intracranial hemorrhage patients, the use of prophylactic anticoagulation was rare due to the risk of exacerbating bleeding.

This study has several limitations. First, it was a retrospective, single-center analysis, which may introduce selection bias. Second, the model lacks external validation. Although the identified independent risk factors were significantly associated with LDVT, causality cannot be established due to the retrospective nature of the study. Future large-scale, multicenter prospective studies are required to confirm these findings and enhance the generalizability of the model.

Conclusion

This study identified hypoalbuminemia and elevated D-dimer levels as independent risk factors for LDVT in critically ill intracranial hemorrhage patients residing in high-altitude regions, while permanent residence at elevations ≥4,500 m appeared to be a protective factor. Based on these variables, a risk prediction model was developed and demonstrated good predictive performance following internal validation. Although the model shows potential clinical utility, further external validation in larger, multicenter cohorts is necessary before it can be widely implemented.

Supplemental Information

Supplemental Information 1 Surgery indications

Supplemental Information 2 Raw Data

We would like to express our sincere appreciation to Danzeng Quzhen for her exceptional input in both the conceptualization and execution of this research.

Additional Information and Declarations

Competing Interests

Author Contributions

Human Ethics

Data Availability

The authors declare there are no competing interests.

Zhuoma Ciren performed the experiments, analyzed the data, prepared figures and/or tables, authored or reviewed drafts of the article, and approved the final draft.

Jianlei Fu analyzed the data, prepared figures and/or tables, authored or reviewed drafts of the article, and approved the final draft.

Guoying Lin conceived and designed the experiments, performed the experiments, authored or reviewed drafts of the article, and approved the final draft.

Qianwei Li performed the experiments, prepared figures and/or tables, and approved the final draft.

Bin Wang conceived and designed the experiments, analyzed the data, prepared figures and/or tables, and approved the final draft.

The following information was supplied relating to ethical approvals (i.e., approving body and any reference numbers):

This research was approved by the Xizang Autonomous Region People’s hospital’s Medical Ethics Committee (ME-TBHP-24-KJ-051).

The following information was supplied regarding data availability:

The raw data are available in the Supplemental File.

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
