# Peer review of "Development of a prediction model for lower limb deep vein thrombosis in critically ill patients after intracranial hemorrhage at high altitude: a retrospective study"

_PeerJ, doi:10.7717/peerj.20245_

## Round 0.1 · original submission · Major Revisions

· Academic Editor

Major Revisions

**Language Note:** The review process has identified that the English language must be improved. PeerJ can provide language editing services - please contact us at [email protected] for pricing (be sure to provide your manuscript number and title). Alternatively, you should make your own arrangements to improve the language quality and provide details in your response letter. – PeerJ Staff

Reviewer 1 ·

Basic reporting

No comment

Experimental design

The authors performed a retrospective analysis of patients with cerebral hemorrhage undergoing surgery who were admitted to a Tibetan hospital. The study was intended to delineate factors associated with formation of lower extremity DVT (L DVT). The introduction succinctly describes the research question and the knowledge gap that the study was intended to address.

Ultimately the authors noted that hypoalbuminemia and elevated D-dimer were independent predictors of L DVT. There are several limitations inherent in the retrospective design. One concern has to do with the possible introduction of bias in patient selection and lab data. The authors do report that screening dopplers were performed daily on all patients, and that those with pre-existing L DVT were excluded. This does help to alleviate concerns about selection bias in some respects, however they do not discuss timing of other laboratory studies, specifically d-dimer, and I note that there are several patients listed in the Raw Data file in whom incomplete lab data were reported. I am concerned that the d-dimer may only have been ordered in patients for whom the index of suspicion was high, which could bias the study. As the authors note, elevated D-dimer has long been used as a screening test for patients with VTE, and it is possible that elevated D-dimer in this population was a result of the DVT, and not a pre-existing risk factor.

Please elaborate on the timing of various lab studies (eg, did every patient have these tests on admission?) and the handling of missing data.

Validity of the findings

Please see above in section 2.

Additionally, please elaborate on the definition of cerebral hemorrhage in this population, and the indication for surgery. Specifically, in many American centers, minimally invasive clot evacuation is currently being performed for nontraumatic lobar cerebral hemorrhage (https://www.nejm.org/doi/full/10.1056/NEJMoa2308440), and for cerebellar hemorrhage with brainstem compression and/or obstructive hydrocephalus, but surgical evacuation is not widely used for deep cerebral hemorrhages (https://www.ahajournals.org/doi/10.1161/STR.0000000000000407).

Relevant questions include:
--Were traumatic hemorrhages excluded? What about those with vascular malformations (AVM, etc.)?
--Were cerebellar hemorrhage excluded? Using the term "cerebral" hemorrhages suggest that this is the case, but the terminology in the literature is somewhat variable. Similarly, does the definition of "cerebral hemorrhage" include subarachnoid and/or subdural hemorrhage?
--When the authors specify that patients were undergoing surgery, what does this entail? For example, does this include external ventricular drainage? Craniotomy with clot evacuation? Non cranial surgery (eg PEG/Trach)? Hemicraniectomy or suboccipital craniectomy?

Reviewer 2 ·

Basic reporting

The manuscript is generally well-written and meets most basic reporting standards. However, a few minor improvements could enhance clarity and professionalism:

Language and Grammar:
Some sentences are overly long or slightly awkward (e.g., "Albumin increases blood volume and maintains a constant colloid osmotic pressure. increases vascular permeability..." on page 11). Rephrasing for conciseness and flow would improve readability.
Ensure consistency in terminology (e.g., "aborigines" vs. "natives" or "long-term residents").

Literature and Context:
The background section effectively covers cerebral hemorrhage and DVT but could briefly clarify why hypoalbuminemia specifically increases LDVT risk (e.g., mechanistic link to blood viscosity or antithrombin III).

Structure and Tables:
Table 2’s layout could be streamlined for better readability (e.g., aligning p-values in a separate column).
Figure legends (e.g., Figure 1) should explicitly state the exclusion criteria (e.g., "9 patients excluded due to pre-ICU thrombosis").

Raw Data:
While raw data are mentioned, confirm accessibility (e.g., supplemental files or repository links) to fully meet journal policy.
These refinements would further polish an already strong manuscript

Experimental design

The experimental design is robust and aligns well with the journal's aims. However, the following points could be clarified or improved:

Ethical Approval:
While the study mentions ethical approval (No. ME-TBHP-24-KJ-051), it would be helpful to briefly state whether consent was obtained for retrospective data use or if the waiver was justified (e.g., anonymization).

Methodological Detail:
Ultrasound Protocol: Specify the frequency of "daily" lower-limb venous ultrasounds (e.g., unilateral/bilateral, operator qualifications, or equipment used) to ensure replicability.
Laboratory Timing: Clarify whether lab values (e.g., D-dimer) were measured at admission, pre-thrombosis, or at fixed intervals, as this affects interpretation.

Confounding Factors:
Address potential confounders (e.g., mobility status, prophylactic anticoagulation use) that could influence LDVT incidence but were not analyzed.

High-Altitude Adaptation:
The protective effect of high-altitude residence is intriguing but lacks mechanistic detail. Briefly hypothesize (e.g., fibrinolytic adaptation) to contextualize findings.

Reproducibility:
The R software version (4.0.3) is noted, but sharing the script/code for the nomogram would enhance transparency.

Conclusion: The study design is rigorous and addresses a clear knowledge gap. Minor clarifications would strengthen methodological transparency.

Validity of the findings

The findings are statistically sound and well-supported by the data, but the following points could enhance validity and clarity:

Causation vs. Correlation:
The study identifies hypoalbuminemia and elevated D-dimer as independent risk factors but does not establish causation. Clarify this limitation in the discussion (e.g., "While associations were significant, causality cannot be inferred due to the retrospective design").

Novelty and Impact:
The protective effect of high-altitude residence is novel, but its mechanism remains speculative. Briefly cite existing literature on hypoxia adaptation (e.g., fibrinolytic activity) to strengthen biological plausibility.

Model Generalizability:
The nomogram’s performance (AUC: 0.815) is strong but requires external validation. Explicitly recommend future multicenter studies to confirm applicability beyond the Tibetan population.

Confounding Adjustments:
Acknowledge unmeasured confounders (e.g., immobility duration, prophylactic anticoagulation) that may influence LDVT risk but were not included in the analysis.

Conclusion Precision:
The conclusions align with results but could temper claims of clinical utility until external validation is performed (e.g., "The model shows promise but requires further validation before widespread adoption").

Overall: The findings are robust and address the research question effectively. Minor revisions would better contextualize limitations and strengthen translational implications.

Additional comments

None.

---

## Round 0.2 · accepted · Accept

· Academic Editor

Accept

Thank you for satisfactorily addressing the concerns of the reviewers. The manuscript is now ready for publication.